# PoET: Pose Estimation Transformer for Single-View, Multi-Object 6D Pose Estimation

**Thomas Jantos**
Control of Networked Systems Group
University of Klagenfurt, Austria
`thomas.jantos@aau.at`

**Mohamed Amin Hamdad**
Infineon Technologies Austria AG
Villach, Austria
`mohamedamin.hamdad@infineon.com`

**Wolfgang Granig**
Infineon Technologies Austria AG
Villach, Austria
`wolfgang.granig@infineon.com`

**Stephan Weiss**
Control of Networked Systems Group
University of Klagenfurt, Austria
`stephan.weiss@aau.at`

**Jan Steinbrener**
Control of Networked Systems Group
University of Klagenfurt, Austria
`jan.steinbrener@aau.at`

**Abstract:** Accurate 6D object pose estimation is an important task for a variety of robotic applications such as grasping or localization. It is a challenging task due to object symmetries, clutter and occlusion, but it becomes more challenging when additional information, such as depth and 3D models, is not provided. We present a transformer-based approach that takes an RGB image as input and predicts a 6D pose for each object in the image. Besides the image, our network does not require any additional information such as depth maps or 3D object models. First, the image is passed through an object detector to generate feature maps and to detect objects. Then, the feature maps are fed into a transformer with the detected bounding boxes as additional information. Afterwards, the output object queries are processed by a separate translation and rotation head. We achieve state-of-the-art results for RGB-only approaches on the challenging YCB-V dataset. We illustrate the suitability of the resulting model as pose sensor for a 6-DoF state estimation task. Code is available at `https://github.com/aau-cns/poet`.

**Keywords:** 6D Pose Estimation, Transformer, Object-Relative Localization

## 1 Introduction

Accurately estimating the 6D pose of objects from RGB images is essential for robotics tasks such as grasping or localization [1, 2]. Grasping tasks require the robot to know the exact position of the object such that it can place its end-effector effectively. In autonomous driving, it is critical that the vehicle has sufficient knowledge of its surroundings including the relative 6D pose of all objects in its vicinity. For unmanned aerial vehicle (UAV) navigation, especially in close proximity to people or infrastructure, realizing precise control depends on the estimation of 6D object poses. In recent years, vision-based 6D pose estimation with deep learning [3] has been on the rise. Approaches differ in terms of input data, network architecture, post processing and number of viewpoints [4, 5, 6, 7]. Observing objects from multiple viewpoints introduces constraints to the pose of objects and improves estimation [7, 8]. Availability of 3D object models allows for an iterative refinement of an initial pose estimate by either iterative closest point (ICP) [9] matching of pointclouds or by matching keypoints with a perspective-n-point (PnP) [10] algorithm. However, these algorithms are computationally demanding. Besides being used for post processing, 3D models can be utilized as an additional input to the network [5]. This may include model keypoints and corresponding features or information about the object such as symmetry axes and planes. Additionally, prior information

6th Conference on Robot Learning (CoRL 2022), Auckland, New Zealand.

about the object class or a depth map corresponding to the input RGB image, which improves the networks ability to estimate the objects' distance to the camera [8], can be passed to the network. Although additional input information can greatly benefit the accuracy of the final estimated pose, apart from requiring a more detailed data base containing accurate 3D models and depth maps corresponding to RGB images, it results in higher computational complexity and thus, longer run time in comparison to the same neural network-based architecture taking only RGB images [8].

While the requirements for real-time performance depend on the specific application, more often the availability of high-quality, detailed 3D models or depth maps is limited. Depending on the type of additional input information, additional sensor hardware is also required, which may not be available during inference. In this work, we focus on a pose estimation framework purely based on RGB images and information provided by a backbone object detector. To the best of our knowledge, we are the first to incorporate global image context information into the pose estimation task by passing multi-scale feature maps to a transformer network and relying only on 2D image information. Our approach does not depend on the number of objects present. Our framework, dubbed PoET (Pose Estimation Transformer), can be used on top of any 2D object detector. We evaluate our approach on the YCB-V [3] dataset and compare our results to state-of-the-art approaches. Finally, we illustrate the suitability of the obtained model as a pose sensor for a 6-DoF state estimation task, where "pose sensor" means the combination of a camera and PoET providing information about the camera pose relative to objects, i.e. its relative position and orientation. Our contributions are the following:

- We present a transformer-based framework that takes a single RGB-image as input, estimates the 6D pose for every object present in the image and can be trained on top of any object detector framework. A detailed ablation study supports our design choices.
- The framework is independent of any additional information which is not contained in the raw RGB image. In particular, it does not depend on depth maps, object symmetries or 3D object models. Hence, our results are achieved without iterative refinement and the whole network can be trained using 3D model independent loss functions.
- We achieve state-of-the-art results on the YCB-V [3] dataset for RGB-only methods and competitive results in comparison to approaches utilizing 3D models.
- We show the feasibility of the resulting model as a pose sensor in a 6-DoF localization task.

The rest of the paper is organized as follows: In Section 2, related work for 6D pose estimation is reviewed. Following the presentation of our method and implementation details in Section 3, the experiments and the corresponding results are discussed in Section 4 including an ablation study investigating our network architecture. Additionally, we illustrate how PoET and its relative 6D pose estimates can be used for localization in Section 4.3. Finally, the limitations are discussed in Section 5 and the paper is concluded in Section 6. We refer the reader to the supplementary material for an extensive ablation study and additional results on the LM-O [11] dataset.

## 2   Related Work

Classical image-based 6D pose estimation approaches can be split into feature-based methods and template-based methods. For the latter, object pose is determined by matching object templates against the input image [12, 13]. While template-based approaches work well on texture-less objects, they are prone to fail in scenarios where objects are occluded. In feature-based methods, local features are extracted from the image and then matched to the 3D model to determine correspondences [14]. Based on these 2D-3D correspondences, the 6D pose of the object can be derived. While these approaches can handle occlusion of objects, they require textures in order to perform the matching. If RGB-D images are available, the additional information provided by the depth can be used to improve the initial pose estimate by iterative refinement [15, 16] such as e.g., ICP [9]. Even though those methods can achieve state-of-the-art performance, they are computationally expensive and the availability of depth maps is not guaranteed.

In recent years, advancements in deep learning for computer vision tasks have been applied to single-view, image-based 6D pose estimation, either to replace components of classical approaches [17, 18, 19, 20], or as end-to-end learned methods, where the 6D pose is directly estimated from the input using convolutional neural networks (CNNs). Xiang et al. [3] proposed with PoseCNN, a CNN-based method to regress the 3D translation and rotation of each object present in the image using

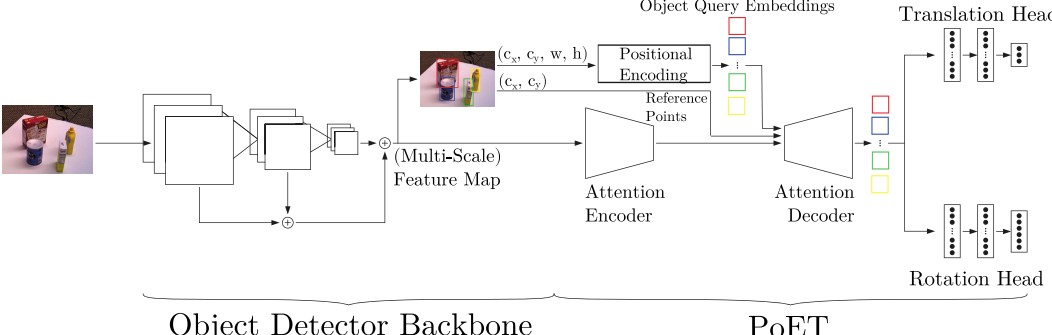

**Figure 1:** Overview of the PoET network architecture for single-view, multi-object 6D pose estimation. Bounding box information for each detected object is passed to the transformer as an object query. Afterwards, for each object query the egocentric 3D translation and 6D rotation [25] is predicted.

an object symmetry-aware loss function. Li et al. [8] proposed a framework that introduces prior knowledge about the object class into the network and perform pose estimation by discretizing the possible translation and rotation values to unique bins resulting in a classification task.

Aside from depth images, using 3D object models for pose estimation yielded promising results [21, 4, 5, 6, 7]. The approaches differ in terms of how the 3D model is used. Kehl et al. [21] and Li et al. [4] predict the 6D pose of objects and then refine the estimated pose. With GDR-Net, Wang et al. [22] are able to integrate PnP into an end-to-end trainable network. Similarly, Li et al. [6] use the 3D model of an object for iterative pose refinement. Given an initial pose estimate, a network is trained to iteratively refine the pose by matching a rendered image created from the 3D model to the original image. Labbé et al. [7] apply this approach to multiple viewpoints of the same scene resulting in improved pose estimation. In yet another approach, Billings and Johnson-Roberson [5] provide the 3D model as an additional input to their network, dubbed SilhoNet. Their network predicts the object silhouette and a 3D translation vector derived from the 2D bounding box position. Based on the former, the 3D rotation of the object is estimated and corrected for silhouette symmetric objects - a significant restriction, as the real error in rotation for symmetric objects can be very large.

Similar to our approach, Amini et al. [23] presented a transformer-based architecture to directly regress the 6D pose for multiple objects contained in a single image. By extending the Detection Transformer [24] with translation and rotation heads, they are able to train the whole network in an end-to-end fashion. However, they require the object 3D model as they use a symmetry aware loss [3]. Our approach does not require any additional information such as depth maps, 3D models or known object symmetries, but instead directly estimates the translation and rotation of objects in the camera coordinate frame from a single RGB image. In contrast to other methods that work with regions of interest for pose estimation, we keep the complete image feature map and provide the regions of interest as an additional input to our transformer network.

## 3 Method

We present a novel transformer-based neural network for the 6D pose estimation task. Taking a single RGB image as its input, the 6D pose of every object detected in the image is predicted simultaneously. After generating (multi-scale) feature maps by passing the image through an object detector, they are processed by a transformer architecture. At the end, translation and rotation are estimated in a decoupled manner. In this section, we first present the general structure of our network. Afterwards, we talk about specific implementation details and data preparation.

### 3.1 Network Architecture

Fig. 1 shows a detailed overview of our network architecture, which consists of three steps. First, the input image is passed through a backbone object detector network. Both the generated (multi-scale) feature maps and the predicted bounding boxes are used in subsequent processing. PoET can be trained on top of any object detector architecture and thus extend a pre-trained object detection framework to include 6D object pose estimation. Second, the (multi-scale) feature maps used for the object detection step are passed to the encoder of a multi-head attention-based transformer.

Our transformer architecture is a modified version of the *Deformable DETR* transformer module proposed by Zhu et al. [26]. In contrast to the original *DETR* [24], the deformable transformer allows to process multi-scale feature maps similar to state-of-the-art object detectors. By only attending to a limited number of feature map keypoints in the decoder, the transformer achieves a higher pass-through rate. Additionally, a deformable transformer shows faster convergence rates than a regular transformer architecture. The main idea behind using a transformer architecture for feature map refinement is to generate features that capture the global information contained in the image. For example, such additional information might be the image location of other objects present or general information of the overall scene extracted from the image.

While the encoder of the deformable transformer is kept unchanged, we modified the decoder to incorporate more information from the object detection step: First, the learned object query embedding is replaced by bounding box information. For each detected object, the bounding box center coordinates $(c_x, c_y)$, the width $w$ and the height $h$ are normalized and then position-encoded [27] to generate the object query embeddings. The embedding dimension $L$ is chosen such that $2 \cdot n_p \cdot L$ equals the hidden dimension $d_h$ of the transformer. In our case, the number of parameters $n_p$ equals to 4. Moreover, the inter-query attention heads ensure that information is properly propagated between the different object queries. Second, the *Deformable DETR* originally only attends to a limited number of keypoints which are randomly sampled around reference points. Instead of predicting reference points from query embeddings by a trainable fully connected layer, we directly feed the normalized center coordinates $(c_x, c_y)$ as the reference points to the decoder. By feeding this additional information to the decoder along with the encoder-refined image feature maps and the inter-query attention heads, the decoder generates new object query embeddings which not only contain local information regarding the object but also global information extracted from the image. Third, the object queries outputted by the transformer are passed through a translation and rotation head. This allows us to simultaneously estimate the pose for multiple objects independent of how many objects are present and which class they belong to. As we approach the pose estimation problem from a global image context perspective by extracting features from the whole image, the network directly estimates the translation and rotation with respect to the camera.

Our translation head is a simple multi-layer perceptron (MLP) with input dimension $d_h$, one hidden layer and output dimension 3. We directly predict the translation $\tilde{t} = (\tilde{t}_x, \tilde{t}_y, \tilde{t}_z)$ with respect to the camera frame. Given the ground-truth translation $t$, our translation head is trained with a simple L2-loss defined as

$$L_t = ||t - \tilde{t}||_2 . \tag{1}$$

The rotation prediction head is identical to the translation head besides the output dimension. For estimating the rotation, we use the 6D rotation representation proposed by Zhou et al. [25] as this representation does not suffer from discontinuities with respect to learning as e.g., quaternions do. Hence, the network predicts a 6-dimensional output vector. Afterwards, the 6D representation is used to determine the estimated rotation matrix $\tilde{R} \in SO(3)$ as described in [25]. The rotation head is trained using a geodesic loss [28] given by

$$L_{rot} = \arccos \frac{1}{2} \left( Tr \left( R\tilde{R}^T \right) - 1 \right) , \tag{2}$$

where $R$ is the ground-truth rotation and $Tr(\cdot)$ is the matrix trace operator. To ensure numerical stability of the loss during training, the argument of the $\arccos$ is clamped between $-1 + \epsilon$ and $1 - \epsilon$, where $\epsilon = 1e - 6$. Our whole network is then trained with a weighted multi-task loss expressed as

$$L = \lambda_t L_t + \lambda_{rot} L_{rot} , \tag{3}$$

where the loss is calculated for each object and then averaged across all objects present across all images in the batch. $\lambda_t$ and $\lambda_{rot}$ are the weighting parameters for the translation and rotation loss respectively. PoET can be trained either class-specific or class-agnostic. In the class-specific case with $n_{cls}$ different classes, the translation and rotation output dimension are changed to $3 \cdot n_{cls}$ and $6 \cdot n_{cls}$, respectively. For each object query, one hypothesis for each class is regressed and the final output is chosen depending on the class predicted for the bounding box. However, no class information is fed into the transformer.

## 3.2   Implementation Details & Data Preparation

While PoET can be trained on top of any object detector, we used Scaled-YOLOv4 [29] as the backbone object detector as it offers a good trade-off between speed and accuracy. An MS-

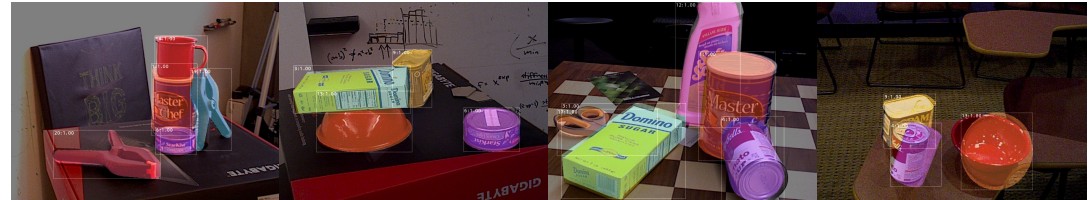

**Figure 2:** Qualitative results of the relative 6-DoF object poses predicted by PoET for the YCB-V dataset.

COCO [30] pre-trained Scaled-YOLOv4 is fine tuned for 10 epochs on the YCB-V dataset for the object detection task. During the training of PoET, the weights of the object detector backbone are frozen.

We implement PoET using PyTorch [31] and train it for 50 epochs using AdamW [32] with a learning rate of $2e-5$ and a batch size of 16. Our best performing network has five encoder and decoder layers, $d_h = 256$, 16 attention heads and a positional embedding dimension of $L = 32$. The network is simultaneously trained for 3D translation and 3D rotation estimation and the weighting parameters are set to $\lambda_t = 2$ and $\lambda_{rot} = 1$, such that both losses are in the same value range.

Given the commonly used data split for YCB-V [3, 5, 6, 7, 8, 33], we train our network on 80 of the available 92 video sequences and reserve the remaining 12 sequences and their challenging keyframes for testing and evaluation. Moreover, we include 80,000 synthetic images generated from 2D projections of the 3D object models provided by the original authors [3]. We refer the reader to the supplementary material, for more details regarding our implementation.

## 4 Results & Experiments

In this section, we present PoET's performance on the YCB-V benchmark dataset for 6D pose estimation. For evaluation, we use the AUC of ADD-S metric [3] and additionally report the average translation and rotation error in $cm$ and degree, respectively, as done by [5]. We compare our results to state-of-the-art, single-view, RGB-based approaches and list the results of other approaches as reported in the corresponding work. We conduct an ablation study on the network architecture and our modifications to the transformer part. The ablation study investigates the influence of the object detector backbone, transformer modifications, data augmentation, network architecture and rotation representation on PoET's performance. While we present here our most important findings, we refer the reader to our supplementary material for additional ablation results. Finally, we illustrate how PoET's relative pose estimates can be used for camera localization.

### 4.1 6D Pose Estimation

Our best performing network is class-specific, consists of 5 encoder and decoder layers with 16 attention heads, and was trained according to Section 3.2. Methods that rely on an object detector to predict regions of interest (ROIs) usually do not elaborate whether and, if so, how multiple predictions, bad predictions and missing predictions are treated within their approach. Therefore, to allow for a fair comparison, we also report the results of PoET and other approaches given ground-truth bounding boxes in addition to the results on predicted bounding boxes. We present representative qualitative results in Fig. 2.

In Table 1 we report our results for the AUC of ADD-S metric. Both for predicted and ground-truth bounding boxes, PoET outperforms (overall and also for most individual classes) other state-of-the-art RGB-based methods that either work on the whole input image [3, 23] or that predict ROIs for pose regression [8]. We also outperform SilhoNet [5] which feeds additional information from the 3D model to its network and reduces predicted rotations by known object symmetries. This shows that PoET and its feature maps containing global image context reduce the need for any additional inputs. This is especially highlighted when comparing the performance of PoET to the ROI-based approaches SilhoNet and MCN [8] in the case where all networks are provided with ground-truth bounding boxes. Only models that explicitly utilize 3D object models during inference, either through PnP during the estimation [22] or by performing iterative refinement after estimating an initial pose [7, 6], achieve slightly better results but this comes at the cost of significantly increased computational complexity and the need for accurate 3D models to be known a priori.

**Table 1:** Comparison with state-of-the-art on YCB-V. We report the AUC of ADD-S. *gt* denotes results achieved with providing ground-truth ROIs to network. The 3D model row indicates how the object model is used: either to calculate the loss function, as an additional input, for symmetry reduction or for PnP or iterative refinement (IR) based pose estimation. (*) denotes symmetric objects. Bold and italic values indicate state-of-the-art results for methods not based on PnP/IR, using ground-truth or predicted ROIs, respectively.

| Method | PoseCNN [3] | SilhoNet [5] | SilhoNet$_{gt}$ | MCN[8] | MCN$_{gt}$ | T6D [23] | PoET$_{gt}$ | PoET | GDR-Net [22] | CosyPose [7] | DeepIM[6] |
|---|---|---|---|---|---|---|---|---|---|---|---|
| 3D Model | Loss | Input + Sym | Input + Sym | 2D | 2D | Loss | 2D | 2D | PnP | IR | IR |
| master chef can | 84.0 | 84.0 | 83.6 | 87.8 | 91.2 | *91.9* | **92.9** | 88.4 | 96.6 | - | 93.1 |
| cracker box | 76.9 | 73.5 | 88.4 | 64.3 | 78.5 | *86.6* | **90.4** | 80.5 | 84.9 | - | 91.0 |
| sugar box | 84.3 | 86.6 | 88.8 | 82.4 | 85.1 | 90.3 | **94.5** | *92.4* | 98.3 | - | 96.2 |
| tomato soup can | 80.9 | 88.7 | 89.4 | 87.9 | 93.3 | 88.9 | **94.0** | *91.4* | 96.1 | - | 92.4 |
| mustard bottle | 90.2 | 89.8 | 91.0 | 92.5 | 91.9 | *94.7* | **94.8** | 91.7 | 99.5 | - | 95.1 |
| tuna fish can | 87.9 | 89.5 | 89.9 | 84.7 | **95.2** | 92.2 | 94.0 | 90.4 | 95.1 | - | 96.1 |
| pudding box | 79.0 | 60.1 | 89.1 | 51.0 | 84.9 | 85.1 | **93.8** | *89.0* | 94.8 | - | 90.7 |
| gelatin box | 87.1 | 92.7 | **94.6** | 86.4 | 92.1 | 86.9 | 92.7 | 91.7 | 95.3 | - | 94.3 |
| potted meat can | 78.5 | 78.8 | 84.8 | 83.1 | 90.8 | 83.5 | **94.1** | *91.2* | 82.9 | - | 86.4 |
| banana | 85.9 | 80.7 | 88.7 | 79.1 | 70.0 | *93.8* | **94.3** | 89.5 | 96.0 | - | 72.3 |
| pitcher base | 76.8 | 91.7 | 91.8 | 84.8 | 91.1 | *92.3* | **94.3** | 91.7 | 98.8 | - | 94.6 |
| bleach cleanser | 71.9 | 73.6 | 72.0 | 76.0 | 86.8 | 83.0 | **92.6** | *85.4* | 94.4 | - | 90.3 |
| bowl* | 69.7 | 79.6 | 72.5 | 76.1 | 85.0 | *91.6* | **92.1** | 90.5 | 84.0 | - | 81.4 |
| mug | 78.0 | 86.8 | 92.1 | *91.4* | 91.9 | 89.8 | **94.1** | *91.4* | 96.9 | - | 91.3 |
| power drill | 72.8 | 56.5 | 82.9 | 76.0 | 87.2 | 88.8 | **94.3** | *88.8* | 91.9 | - | 92.3 |
| wood block* | 65.8 | 66.2 | 79.2 | 54.0 | 87.2 | *90.7* | **92.0** | 75.7 | 77.3 | - | 81.9 |
| scissors | 56.2 | 49.1 | 78.3 | 71.6 | 80.2 | *83.0* | **92.5** | 75.2 | 68.4 | - | 75.4 |
| large marker | 71.4 | 75.0 | **83.1** | 60.1 | 66.4 | 74.9 | 81.6 | *81.2* | 87.4 | - | 86.2 |
| large clamp* | 49.9 | 69.2 | 84.5 | 66.8 | 86.5 | 78.3 | **95.7** | *88.6* | 69.3 | - | 74.3 |
| extra large clamp* | 47.0 | 72.3 | 88.4 | 61.1 | 79.5 | 54.7 | **96.0** | *83.5* | 73.6 | - | 73.2 |
| foam brick* | 87.8 | 77.9 | 88.4 | 60.9 | 79.2 | *89.9* | **89.7** | 81.3 | 90.4 | - | 81.9 |
| All | 75.9 | 79.6 | 85.8 | 75.1 | 86.9 | 86.2 | **92.8** | *87.1* | 89.1 | 89.8 | 88.1 |

The improved performance of PoET compared to other RGB-based models in terms of the ADD-S score is due to a better estimate of the 3D translation as can be seen in Table 2. Again, ground-truth-based results are also presented. In contrast to directly estimating the 3D translation like PoET, PoseCNN as well as SilhoNet determine the 3D translation by estimating the depth and center pixel coordinates of an object and then reprojecting them using the known camera intrinsic parameters, which is considered the easier task to learn [3]. MCN treats the translation estimation as a classification problem by binning the translation space. In addition to the global image information provided by the multi-scale feature maps, our approach also learns to model the camera intrinsics, which results in a more accurate estimation of translation.

For 3D rotation estimation, we achieve the same average error as regular PoseCNN. Not surprisingly, networks that reduce the possible rotation space based on known object symmetries achieve a better result but with limited applicability to real-world scenarios. The mean average 3D rotation error is shown in Table 2. The influence of reducing the rotation by geometrical symmetries (†) is highlighted for PoseCNN. Since PoET makes use of the full multi-scale RGB feature maps, it outperforms SilhoNet for objects that have no symmetries in the silhouette space and only performs significantly worse for objects with rotational symmetries around an axis in 3D space, but without requiring a priori knowledge about the 3D shape of objects or restricting the rotation space based on symmetries. The main source for rotational errors are objects with rotational symmetries around one axis (master chef can, tomato soup can, tuna fish can). We have investigated the axes of our rotation errors and compared them to the symmetry axes for symmetric objects. The average tilt of rotation error axes with respect to symmetry axes across all test images and symmetric objects is only 15 degrees. If we ignore rotational errors about symmetry axes, our average rotation error reduces to 11.24 degrees, outperforming all RGB-only competitors, see Table 2. We refer the reader to the supplementary material for additional ablation experiments as well as PoET's performance for the stricter BOP[33] and AUC of ADD [3] metrics and for the LM-O benchmark dataset [11].

## 4.2 Ablation Study

The ablation study investigates the influence of different components on the performance of our PoET framework. By assuming a perfect object detector that provides ground truth bounding boxes to PoET, we ensure that the evaluation includes every object present in the image even for those which might be not detected by the object detector. During the ablation study, we focus on the AUC of ADD/ADD-S metric, the average translation error and rotation error. All results reported in this section are for the same hyperparameter configuration as described in Section 3.2, the same fixed seed and trained for the same number of epochs. The final results are summarized in Table 3. We compare our best performing network from Section 4.1 (`Baseline`) to a class-agnostic version (`Agnostic`) and a network with less layers (`Small`). Finally, we investigate the influence of the integration of bounding box information into the transformer on the performance of PoET. We train PoET with learnable reference points (`RP`), trainable query embeddings (`Q`) or the combination of

**Table 2:** Comparison of average translation in cm and rotation error in degrees on YCB-V. Bold and italic values indicate the state-of-the-art for results, when working on ground-truh or predicted ROIs respectively. † indicates that the rotation predictions are reduced by geometric symmetries as described in [5].

| Method | PoseCNN[3] | SilhoNet[5] | SilhoNet$_{gt}$ | PoET$_{gt}$ | PoET | PoseCNN | PoseCNN† | SilhoNet† | SilhoNet$_{gt}$ | PoET$_{gt}$ | PoET |
|---|---|---|---|---|---|---|---|---|---|---|---|
| | *Translation Error [cm]* | | | | | *Rotation Error [°]* | | | | | |
| master chef can | 3.29 | 3.02 | 3.14 | **1.37** | 2.26 | 50.7 | 7.57 | *1.21* | **1.11** | 89.25 | 80.12 |
| cracker box | 4.02 | 5.24 | 2.38 | **1.48** | 3.14 | *19.69* | 19.69 | 19.86 | **9.53** | 9.68 | 21.87 |
| sugar box | 3.06 | 2.10 | 1.67 | **0.94** | 1.42 | 9.29 | 9.29 | 12.28 | 11.50 | **3.95** | *4.40* |
| tomato soup can | 3.02 | 2.40 | 2.24 | **1.09** | 1.62 | 18.23 | 8.40 | *1.91* | **1.82** | 50.97 | 49.29 |
| mustard bottle | 1.72 | 1.65 | 1.41 | **0.94** | 1.42 | 9.94 | 9.59 | *5.78* | **5.07** | 23.71 | 27.73 |
| tuna fish can | 2.41 | *1.57* | 1.49 | **0.95** | 1.79 | 32.80 | 12.74 | *1.46* | **1.50** | 60.30 | 63.72 |
| pudding box | 3.69 | 7.15 | 1.91 | **1.01** | 1.94 | 10.20 | 10.20 | 20.95 | 18.39 | **6.36** | 6.87 |
| gelatin box | 2.49 | *1.09* | **0.79** | 1.20 | 1.41 | *5.25* | 5.25 | 12.52 | 8.48 | **6.69** | 7.19 |
| potted meat can | 3.65 | 4.30 | 2.74 | **1.13** | 1.75 | 28.67 | 19.74 | 7.27 | 10.93 | **5.06** | 6.75 |
| banana | 2.43 | 4.12 | 2.59 | **1.06** | 1.95 | *15.48* | 15.48 | 16.29 | **5.70** | 7.90 | 20.40 |
| pitcher base | 4.43 | *1.31* | 1.29 | **0.95** | 1.55 | 11.98 | 11.98 | 6.64 | **6.61** | 7.51 | 8.04 |
| bleach cleanser | 4.86 | 3.60 | 3.99 | **1.09** | 2.47 | *20.85* | 20.85 | 51.28 | 48.42 | **16.32** | 21.93 |
| bowl* | 5.23 | 3.30 | 4.08 | **1.51** | 1.76 | 75.53 | 75.53 | 49.95 | 53.95 | **16.06** | *25.71* |
| mug | 4.00 | 2.61 | 1.43 | **1.28** | 1.85 | 19.44 | 19.44 | 18.14 | 7.02 | **3.86** | *5.59* |
| power drill | 4.59 | 6.77 | 3.19 | **0.98** | 2.29 | 9.91 | 9.91 | 30.54 | 10.66 | **5.92** | 6.45 |
| wood block* | 6.34 | 5.59 | 3.23 | **1.41** | 4.75 | 23.63 | 23.63 | 25.52 | 23.23 | **5.88** | *14.32* |
| scissors | 6.40 | 9.91 | 2.59 | **1.38** | 3.72 | 43.98 | 43.98 | 155.53 | 154.82 | **3.19** | *6.27* |
| large marker | 3.89 | 3.24 | **2.31** | 2.68 | 2.75 | 92.44 | 13.59 | *10.44* | **10.72** | 24.95 | 25.91 |
| large clamp* | 9.79 | 6.27 | 3.51 | **0.98** | 2.33 | 38.12 | 38.12 | *3.54* | 6.03 | **2.61** | 4.88 |
| extra large clamp* | 8.36 | 4.86 | 2.12 | **0.91** | 3.10 | 34.18 | 34.18 | 29.18 | 7.30 | **2.38** | *26.01* |
| foam brick* | *2.48* | 3.98 | 2.31 | **1.90** | 3.42 | 22.67 | 22.67 | *13.84* | **17.36** | 37.20 | 36.34 |
| All | 4.16 | 3.49 | 2.45 | **1.20** | 2.12 | 27.79 | 17.82 | *16.04* | **13.48** | 23.65 | 27.26 |

**Table 3:** Ablation study results of PoET on YCB-V. We report the AUC of ADD/-S and the average translation and rotation error. Ablation of class mode (`Agnostic`), network size (`Small`) and transformer modifications (`RP`, `Q`, `RP + Q`). The exact meaning of the tags are described in the text.

| Metric | Baseline | Agnostic | Small | RP | Q | RP + Q |
|---|---|---|---|---|---|---|
| AUC of ADD-S | **92.8** | 88.9 | 91.8 | 87.6 | 82.2 | 42.0 |
| AUC of ADD | **80.8** | 73.2 | 78.1 | 66.8 | 59.5 | 12.2 |
| Avg. T. Error [cm] | **1.20** | 1.95 | 1.48 | 1.92 | 2.99 | 9.06 |
| Avg. Rot. Error [°] | **23.65** | 24.92 | 25.64 | 37.31 | 35.39 | 74.26 |

both (`RP + Q`). In all three cases the performance is reduced in comparison to a version of PoET that uses bounding box information to generate query embeddings and reference points. This shows that providing a transformer with bounding box information can greatly benefit its training process leading to improved performance for the same training duration. We kindly refer the reader to the supplementary material for an in-depth analysis and discussion of the ablation study.

## 4.3 Localization

PoET is well suited for vision-based object-relative localization. A transformation between coordinate frames $A$ and $B$ expressed in coordinate frame $C$ is fully defined by the translation $^C t_{ab}$ and the rotation $R_{ab}$. Given a set of landmarks with known ground-truth pose $(R^i_{wo}, t^i_{wo})$, a single frame that captures at least one of those landmarks can be used in combination with PoET to localize the camera by estimating the relative pose $(\tilde{R}^i_{co}, \tilde{t}^i_{co})$ to all landmarks present in the frame. For each landmark, the estimated camera pose $(\tilde{R}^i_{wc}, \tilde{t}^i_{wc})$ can be determined by

$$\tilde{R}^i_{wc} = R^i_{wo} \tilde{R}^{i^T}_{co} \quad \text{and} \quad {}^W \tilde{t}^i_{wc} = {}^W t^i_{wo} - R^i_{wo} \tilde{R}^{i^T}_{co} {}^C \tilde{t}^i_{co}. \tag{4}$$

The final camera pose $(\tilde{R}_{wc}, \tilde{t}_{wc})$ can then be determined by taking the average over all landmarks present in the image. YCB-V's test sequences offer 12 different camera trajectories and each with a different constellation of multiple objects serving as landmarks. For each individual frame we estimate the camera pose and compare it to the ground-truth. In Fig. 3 we show an example trajectory for a single sequence. For further examples we refer the reader to the supplementary material.

We compare three different approaches: using all detected objects (`all`), perform simple outlier rejection by taking and choosing the hypothesis the majority of objects agree on (`out`) or by incorporating the camera pose estimate from the previous frame into the outlier rejection in cases with multiple hypotheses having the same number of votes (`prev`). For the sake of comparison, we also calculate the camera pose by choosing the estimated camera pose being closest to the current ground-truth camera pose (`best`). Moreover, we differentiate between PoET being provided either object detections from the backbone (`bb`) or ground-truth bounding box information (`gt`). We calculate the sample error mean and standard deviation across all frames for the position and attitude and summarize them in Table 4. Simple outlier rejection greatly benefits the localization error. This is due to wrong relative object pose estimates having a large impact on the final estimated camera pose when taking a simple average. Incorporating the camera pose estimate from the previous frame only slightly improves the localization as it only helps in multi-hypothesis

cases. In comparison to the estimated trajectories using the best estimated camera pose, our outlier rejection based trajectories perform worse with around factor two. Nonetheless, the localization results when considering only individual frames are remarkable. The difference in performance between backbone and ground-truth detections is due to the backbone potentially missing an object or assigning a wrong class and thus throwing off the estimate.

To further motivate the use of PoET as a pose sensor in mobile robotics, we have integrated it in a state-of-the-art sensor fusion framework [34] and performed state estimation experiments with YCB-V objects in our motion capture room using inertial data for propagation and PoET for the pose update. For a representative trajectory, the average error in rotation was (roll, pitch, yaw) = (4.0, 6.9, 14.8) degrees and in translation (x, y, z) = (0.084, 0.179, 0.032) meters. The performance for our real data is only slightly worse compared to the performance on the benchmark dataset, even though we were using a completely different camera, than the one used to record the original dataset, and the objects have slightly changed in appearance in comparison to the original YCB objects. Nonetheless, PoET is able to provide sufficiently accurate pose data. Summarizing, PoET achieves sufficient localization accuracy such that it can be used as a pose sensor for robotics.

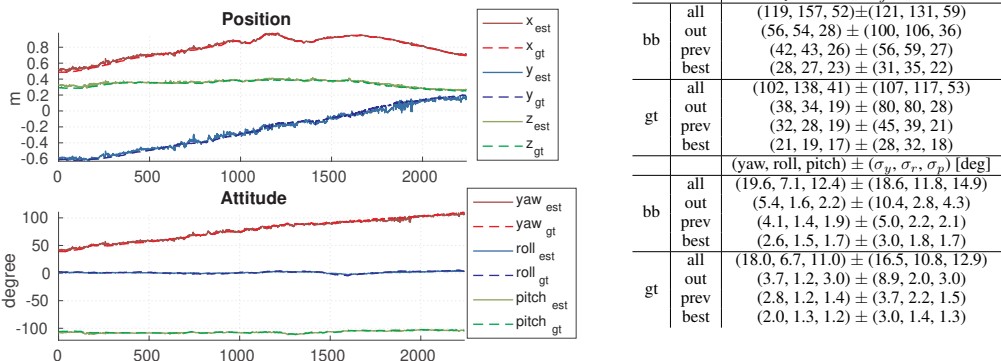

**Figure 3:** Example trajectory of a single sequence for `gt` and `out`. We plot the ground-truth and estimated position and attitude.

| | | $(x, y, z) \pm (\sigma_x, \sigma_y, \sigma_z)$ [mm] |
|---|---|---|
| bb | all | (119, 157, 52)±(121, 131, 59) |
| | out | (56, 54, 28) ± (100, 106, 36) |
| | prev | (42, 43, 26) ± (56, 59, 27) |
| | best | (28, 27, 23) ± (31, 35, 22) |
| gt | all | (102, 138, 41) ± (107, 117, 53) |
| | out | (38, 34, 19) ± (80, 80, 28) |
| | prev | (32, 28, 19) ± (45, 39, 21) |
| | best | (21, 19, 17) ± (28, 32, 18) |
| | | (yaw, roll, pitch) $\pm (\sigma_y, \sigma_r, \sigma_p)$ [deg] |
| bb | all | (19.6, 7.1, 12.4) ± (18.6, 11.8, 14.9) |
| | out | (5.4, 1.6, 2.2) ± (10.4, 2.8, 4.3) |
| | prev | (4.1, 1.4, 1.9) ± (5.0, 2.2, 2.1) |
| | best | (2.6, 1.5, 1.7) ± (3.0, 1.8, 1.7) |
| gt | all | (18.0, 6.7, 11.0) ± (16.5, 10.8, 12.9) |
| | out | (3.7, 1.2, 3.0) ± (8.9, 2.0, 3.0) |
| | prev | (2.8, 1.2, 1.4) ± (3.7, 2.2, 1.5) |
| | best | (2.0, 1.3, 1.2) ± (3.0, 1.4, 1.3) |

**Table 4:** Sample error mean and standard deviation of camera pose across all 12 test sequence frames. Position and attitude are reported in $mm$ and degree.

## 5   Limitations

In Section 4.1 it was discussed that PoET is outperformed by methods utilizing 3D object information for rotation estimation in particular for objects that have rotation-symmetric silhouettes. Wrongly estimated rotations lead to the assumption that the camera views the objects from a different angle resulting in wrong hypotheses for the localization task as discussed in Section 4.3. Moreover, the low resolution of the images in the YCB-V dataset means that RGB textures are not as dominant, especially when augmentation is used. This is the main reason why PoET has difficulties to estimate silhouette rotation-symmetric objects that are not symmetric in RGB space, see e.g. Table 2.

## 6   Conclusion

In this work we presented a novel, transformer-based framework for multi-object 6D pose estimation. It can be used on top of any object detector and the only input required is a single RGB image. The image is passed through an object detector backbone to create (multi-scale) image feature maps and to detect objects. Bounding box information of detected objects is fed into the transformer decoder which improves the learning. By taking the whole image into consideration during the estimation process, our framework does not rely on any additional information. We outperform other RGB-based methods by a wide margin on the YCB-V dataset. This is especially important for scenarios where no detailed 3D models or prior object information is available, or where computational efficiency is required and thus any input besides the RGB image has to be dropped. Moreover, we highlighted how PoET's relative 6D object pose estimation can be used as a pose sensor for robot localization tasks.

**Acknowledgments**

This work was supported by the Federal Ministry for Climate Action, Environment, Energy, Mobility, Innovation and Technology (BMK) under the grant agreement 881082 (MUKISANO).

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
