# OpenReview forum: "PoET: Pose Estimation Transformer for Single-View, Multi-Object 6D Pose Estimation"
_robot-learning.org/CoRL/2022/Conference — CoRL 2022 Poster_

### Official Review · Reviewer_jp2U · 2022-07-20

**Originality:** Very Good
**Technical Quality:** Very Good
**Clarity Of Presentation:** Fair
**Impact:** 4

**Recommendation:**

Weak Accept: I recommend accepting the paper, but will not argue for my recommendation if the majority of other reviewers have a different opinion.

**Summary:**

The paper presents a framework for single-view, multi-object 6-D pose estimation. The method is based on an object detector backbone, which receives a single RGB image as input, and a modified version of a Deformable DETR Transformer, which is provided with multi-scale features extracted by the object detector, as well as the center coordinates and extent of the bounding-box predicted by the latter. The "object queries" returned by the Transformer are passed through two separate MLP to regress a translation and rotation vector for each detected object. The object detector is finetuned from a pretrained model, while the Transformer and MLPs are trained end-to-end with pose supervision  and receiving ground-truth bounding box and object class at training time. The approach is novel and achieves state-of-the-art results on the YCB-V and LM-O datasets for RGB-only methods. The method is also demonstrated for the localization of a camera against a set of landmarks on scenes from a dataset.

**Issues:**

# Points to clarify
- L39: What does "_detailed_ data _base_" mean exactly? Also, this and the previous statements in the paragraph should be supported by references
- L40: "architecture": Is this referring only to network-based solutions? If so, it should be specified
- "pose sensor" (L52, 62, 299, 320): What exactly is meant by this? "Sensor" commonly refers to a piece of hardware
- L74: [7] does not assume 3D models
- L77: [9] does not perform 6D pose estimation, nor does it use a 3D model
- L91: What does "binning the translation and rotation" mean?
- L142: While represented with $\gamma$, the encoding from NeRF is never referred to in [28] as "gamma encoding"
- Eq. 2-3: The text should mention what R, $\lambda_t$ and $\lambda_{rot}$ are
- L176: Missing meaning of $n_{cls}$​. This part could be clearer in general, e.g., specify that one pose hypothesis is regressed for each class
- L193: How exactly is the match performed?
- Tab. 3: How can the large discrepancy on "scissors" be interpreted?
- L263: PoseCNN -> SilhoNet?
- Eq. 4: Why "W" in the top-left corner of translation only? Why "wo" subscripts in the RHS of translation and "wc" in the LHS?
- L294-295: How can this large discrepancy be interpreted?
- L308: Can this be supported with results?
- Suppl.
  - Sec. 2: Why does RP + Q perform much worse?
  - L67-69: If Jitter is defined in the text as having $\sigma=0.02$, how can it be evaluated for different sigmas in Fig. 1? Clarify that it is different from Tab. 1
  - L83: How can this be supported?
  - L131, Tab. 6: What does GT mean for YOLO/R-CNN? The GT bounding boxes do not come from a network
  - L138: Is Mask R-CNN or Faster R-CNN used?
  - Fig. 5: What does "out" mean?
  - L178: [13] -> [12]?

# Visualization
- Fig. 1: Unusual style: Very thin lines, not clear what the dots in the right part represent. Writings at the bottom are crucial for understanding, but barely visible. Almost all the text (e.g., $c_x, c_y$) is too small to be read without zoom
- Table 2-4, Fig. 2, Suppl.: Fig 3.: Numbers, text and lines are way too small to be read without zoom
- Suppl.
  - Tab. 1: Symbols in the column headers are not defined in the caption. At least a sentence mentioning that these are defined in the text should be inserted
  - Bold should be used in all Tables to highlight results, particularly in Tab. 4-5, in which Baseline does not always achieve the lowest error
- Inconsistent fonts across figures (cf., e.g., Fig. 1 and Suppl. Fig. 1)

# Typos/formatting
- Dashes in multi-word adjectives
  - state of the art -> state-of-the-art: L12, 60, 131, 223, 239. Suppl.: L71, 132, 140, 177, 181, 186, 204, Caption Tab. 6, 9
  - 6 DoF -> 6-DoF: L14
  - RGB only -> RGB-only: L60
  - pass through -> pass-through: L133
  - encoder refined -> encoder-refined: L150
  - ground truth -> ground-truth: L160, 192, 195, 235, 246, 273, 287, 297, Caption Tab. 1. Suppl.: L58, 59, 131, 135, 155, 170, 185, 197, Caption Fig. 1, 4
  - 6 dimensional -> 6-dimensional: L165
  - class specific/agnostic -> class-specific/agnostic: L172, 173. Suppl.: L19
  - real world -> real-world: L261
  - rotation symmetric -> rotation-symmetric: L303, 308
- L31: an -> a
- Missing commas after the following words
  - L36: "image"
  - L49: "framework", "Transformer)"
  - L88: "PoseCNN"
  - L101: Before "dubbed"
  - L111: "symmetries"
  - L123: "architecture"
  - L192: "disregarded"
  - L194: "inference"
  - L251: "translation"
  - L257: "intrinsics"
  - "e.g." -> ", e.g., ": L81, 164. Suppl.: L120
- L37: Missing apostrophe after "objects"
- L38: apart of -> apart from
- L43: What does "it" refer to?
- L46: to the -> into the
- L78: this -> these
- L98: takes [...] applies -> take [...] apply
- L120: section we, -> section, we
- L139: ";" at the end of paragraph -> ", as follows."?
- positional encoded -> positionally-encoded: L142, Suppl.: L109
- Exp. notation: $e$ -> $\textrm{e}$, else it is Euler's number: L169, Suppl.: Caption Fig 6
- L214: Remove comma after "though"
- Missing period at the end of caption: Table 1-3. Suppl.: Table 1-9
- L233: ROI -> ROIs
- L243: reduces -> reduce
- L254: "it" -> "them", or rephrase
- L285: hypothesis -> hypotheses
- Missing venue for ref. [5]
- Ref. [27] and [33]: Replace arXiv with ICLR 2021/2019 resp.
- Format of [32] is not uniform with other references (e.g., URL)
- Suppl.
  - L3: work -> works
  - L17: is -> are
  - Fig. 1: "AuC" -> "AUC" for consistency with text
  - L140: detector -> detectors
  - Caption Tab. 7: perfromance -> performance
  - Caption Fig. 4: Remove "as"
  - L199: PNP -> PnP

# Language suggestions
- L32-33: "Besides using 3D object models [...], they can be" -> "Besides being used for post processing, 3D models can be"
- L89: ". Moreover, [...] employ" -> ", which makes use of"
- L100: "as they" -> "who"
- L129: "attention-based" is redundant. One could rather specify "_vision_ transformer"
- Suppl.
  - L23: "layers," -> "layers;"
  - L216: "to further improve" -> "on further improving"

**Quality Of The Limitations Section:**

Limitations are addressed clearly

**Reviewer Expertise:**

4: The reviewer is confident but not absolutely certain that the evaluation is correct

**Robotics Focus:**

Highly relevant to robotics but no hardware experiments

**Strengths And Weaknesses:**

The ideas and the architecture proposed by the paper are mostly novel and provide interesting insights about the use of (Deformable DETR) Transformer networks for multi-object pose estimation. The experiments are thorough and extensive and the comparisons with previous work fair (e.g., all the baselines use synthetic data for augmentation during training, as the proposed method). The approach achieves state-of-the-art on the evaluated datasets for RGB-only methods, and runs at real-time rates. The supplementary material provides a large number of additional results and analyses.

However, a substantial improvement in the quality of the presentation is required. In particular:

- The quality of the visualizations is in my opinion not yet at the level of an international conference: Many writings in the figures and tables have an extremely small font size, and are hard to read even after zooming to 300%. Figure 1, as well as the line plots (Fig. 2, Suppl. Fig. 3. and 5) could be restyled to appear more professional, by, e.g.: making the font and its size uniform across the visual elements and w.r.t. the other figures (and ideally the text), making the colors more distinctive in the line plots for overlapping lines (so as to increase readability), and explaining the meaning of the legend and the other visual elements (e.g., the MLPs in Fig. 1).
- A large amount of typos and formatting issues are currently present (see "Issues -> Typos/formatting").
- Language is at times articulated or inaccurate (see "Issues -> Language suggestions" and partly "Issues -> Points to clarify")
- Occasionally, symbols are used, either in the equations or in Tables, without being introduced in the previous text or caption respectively.

Citations of related work sometimes lack preciseness (e.g., papers cited as examples of a class of method, but not actually matching the features mentioned in the text for these methods). A number of general statements made in the introduction and to provide context for the related work would need to be supported by references.

Finally, while for the "pose sensor" experiments results are shown that could be applied in a robotic scenario for camera localization, no actual deployment on hardware is demonstrated.

**Summary Of Recommendation:**

Overall, I recommend accepting the paper, given that the proposed ideas are mostly novel, that the task investigated is of concrete relevance for robotics and that the results demonstrated, as well as the running speed of the algorithm, could make the method particularly suitable for real-world robotic applications. However, a non-negligible amount of editing should be performed to address the issues detailed in the "Issues" section.

---

> ### Author Response · Authors · 2022-08-26
> **Reply to Reviewer jp2U**
>
> We appreciate the important points raised by the reviewer.
>  * We apologize for the insufficient quality of the visualizations and are very grateful for the detailed summary of typos and language issues. We will update our manuscript accordingly.
>  * Please find our answers to the raised “points to clarify” below. We will update our manuscript accordingly.
>    - A “detailed database” is one that contains an accurate 3D model and/or depth maps corresponding to the RGB images. We will add references as suggested.
>    - Yes, it is referring to network-based solutions.
>    - In this context, “pose sensor” is an abstract term for a hardware and /or software sensor that gives information about the pose (i.e. position and attitude) of the robot. In our case, the pose sensor consists of an RGB camera together with PoET providing 3D position and 3D attitude information.
>    - True, we listed [7] as an example for template matching. We will reword this sentence.
>    - [9] will be removed
>    - "Binning the translation and the rotation" refers to discretizing the possible translation and rotation values to unique bins resulting in a classification task.
>    - We agree that [28] does not refer to NeRF as gamma encoding. We will change that.
>    - R, $\lambda_t$ and $\lambda_{rot}$ will be clarified.
>    - We like the idea of clarifying that for each object query one hypothesis for each class is regressed.
>    - We will clarify that we use an adjusted Hungarian Matcher similar to [27].
>    - The discrepancy on scissors is since SilhoNet adjusts the rotations for symmetric objects. However, the scissor is not defined as symmetric in the original YCB-V dataset, even though it is quite symmetric. This shows that SilhoNet's rotation estimation relies heavily on the symmetry correction, while PoET can deal with it.
>    - PoseCNN is correct here. In Table 3, indicated by the dagger, the results are reported also for PoseCNN with rotations adjusted for object symmetries. However, the influence of reducing the rotation by geometric symmetries is shown for both.
>    - The ${}^Wt_{wc}$ is the translation of the camera c to the world origin w expressed in the world frame W, while $R_{wc}$ is the rotation of the camera expressed in the world frame. The latter is unambiguous, so we do not need the LHS superscript W for rotations. Likewise, $R_{co}$ is the rotation of object o expressed in the camera frame c. We will better explain the notation in the manuscript. In Eq.4, we want to express the desired $t_{wc}$ by a chain of known (position of landmark, measured pose) coordinate frame transformations. We accidentally dropped a term from the translation equation and will add it. The full translation equation would be ${}^Wt_{wc} = {}^Wt_{wo} - R_{wo} * R_{co}^T * {}^Ct_{co}$
>    - The discrepancy can be explained that for "best" we use a single measurement which is closest to the ground truth pose. In our outlier rejection approach, there can still be multiple objects where the average of the estimated poses is further off from the ground truth.
>    - Table 3 shows the results of the rotation error for the different classes. Objects with rotational symmetry in silhouette space have higher rotational error. We hypothesize that with better RGB textures on the training and testing data, PoET would be able to predict the rotation around the symmetry axes with higher accuracy. A detailed analysis is left for future work.
>    - Suppl.
>      - For the ablation study, we train all the networks for the same duration. In the case of RP+Q the network must learn to identify interesting image regions by itself. This leads to drastically reduced performance for the same training regimen.
>      - For Tab. 1 and Fig. 1, PoET was trained with a jitter of sigma = 0.02, but the evaluation for Tab.1 was still conducted with ground truth bounding boxes to make the comparison of the ablation networks independent of the quality of the object detector. For Figure 1, PoET is evaluated with the jittered bounding boxes. We will clarify this in the manuscript.
>      - This statement is supported by Tab. 2, 4 and 5. When comparing the achieved ADD-S scores of the ablation networks to their average translation and rotation errors, we see that the better the translation estimation is, the better the ADD-S score is, while the rotation seems to have a slightly smaller influence (see e.g. Agnostic and Small)
>      - We will clarify that the tags YOLO and R-CNN refers to PoET trained on top of either one of the object detector backbones. We use the ground truth bounding boxes, but then utilize the feature maps of the respective object detector.
>      - L 138 must be changed to Mask R-CNN, we reworked that section and forgot to change that instance.
>      - "out" refers to the same description as in the main paper. It is the combination of outlier rejection and combination of multiple objects pose hypothesis.
>      - The reference in line 178 must be changed to 12. Thanks for pointing that out.

---

### Official Review · Reviewer_pB8h · 2022-07-31

**Originality:** Good
**Technical Quality:** Good
**Clarity Of Presentation:** Very Good
**Impact:** 4

**Recommendation:**

Weak Reject: I recommend rejecting the paper, but will not argue for my recommendation if the majority of other reviewers have a different opinion.

**Summary:**

This paper tackles the problem of 6D pose estimation object instances. The method is pretty standard. It is a transformer based architecture that takes in the features at different scales of the object detector as the input along side the positionally encoded information for each detection and predicts rotation and translation for each object. The method is evaluated on YCB-V dataset. It achieves decent (but not SOTA) on pose estimation.


**Issues:**

See weaknesses above.

**Quality Of The Limitations Section:**

Limitations are addressed clearly

**Reviewer Expertise:**

5: The reviewer is absolutely certain that the evaluation is correct and very familiar with the relevant literature

**Robotics Focus:**

Highly relevant to robotics but no hardware experiments

**Strengths And Weaknesses:**

Weaknesses:
- Why both bbox coordinates and object centers are provided to transformer. What is the intuition behind having bbox info as object codes (as opposed to fixed learned object codes in other works) + providing center as the reference point? How would the results change if for example, object queries are fixed and learned and only reference points are provided?
- Backbone effect: the method takes as input the features of Yolo. I would like to see if the same results hold if the detector is changed or the method is tied to YOLO object detector.
- Loose connection to robotics application: I would like to see an experiment where the proposed method is used to estimate the pose of an object and then robot manipulate the object.


**Summary Of Recommendation:**

I am more leaning toward the borderline and open to change based on authors' response to my comments.

---

> ### Author Response · Authors · 2022-08-26
> **Reply to Reviewer pB8h**
>
> We are grateful to the reviewer for the feedback.
>
> * The reason for providing the transformer bounding box information in form of object queries as well as reference points is to speed up the learning. The speed-up achieved by Deformable-DETR is due to only sampling around certain reference points, which are points of interest. Points of interest, namely bounding box center coordinates, are provided by the object detector backbone and hence our aim was to utilize this information. Moreover, the bounding box dimension and position already encodes useful information about the object, its dimension and thus translation. Therefore, we want to provide this information to the transformer. We investigate the effect of learnable object queries and/or reference points in the supplementary material (see last three columns in Table 2 of supplementary). This analysis shows that our design choices significantly improve the performance. In fact, learning both queries and reference points as in the standard architecture results in an AUC of ADD-S metric of only 42.0, less than half of what our baseline model achieves given that both networks were trained for the same number of epochs. These findings will be moved to the main part of the manuscript to better motivate our design choices and support our contributions.
> * A detailed analysis of the influence of the object detector backbone on the results of PoET can be found in the supplementary material. We have evaluated both with an existing different object detector (Mask R-CNN, see section 4 in supplementary), and with a hypothetical object detector by simulating different degrees of bounding box jitter (see page 2 and 3 of supplementary). Training PoET with Mask R-CNN instead of Yolo yields almost the same result (AUC of ADD-S of 86.1 instead of 87.1, see Table 6 in supplementary). We agree that these are important aspects and hence we will add a short summary containing the most important findings of our ablation study to the camera-ready version of our paper.
> * Moreover, we want to provide additional results to show the connection to robotics application. While robot manipulation is one task that benefits from relative pose estimation, another important task is object relative localization. In particular for mobile robotics such as drones, being able to estimate it’s own position accurately with respect to objects of interest is crucial for many applications, e.g., the autonomous inspection of critical infrastructure or automated inventorying. To further illustrate the suitability of PoET as a pose sensor for such semantic navigation tasks, we recorded real data using objects from the YCB-V dataset in our motion capture room and integrated PoET into a modular sensor fusion framework for mobile robotics. This framework fuses our relative pose estimates from PoET with inertial data from the UAV platformdata to perform trajectory estimation and landmark pose estimation with respect to the world frame. For a representative trajectory, we have an average translation error of (x, y, z) = (0.084, 0.179, 0.032) meters and orientation error of (roll, pitch, yaw) = (4.0, 6.9, 14.8). The performance for our real data is only slightly worse compared to the performance on the benchmark dataset, even though we were using a completely different camera, than the one used to record the original dataset, and the objects have slightly changed in appearance in comparison to the original YCB objects. Nonetheless, PoET is able to provide sufficiently accurate pose data. This new experiment will be added to the manuscript.

---

### Official Review · Reviewer_v6yy · 2022-08-01

**Originality:** Good
**Technical Quality:** Very Good
**Clarity Of Presentation:** Excellent
**Impact:** 3

**Recommendation:**

Weak Reject: I recommend rejecting the paper, but will not argue for my recommendation if the majority of other reviewers have a different opinion.

**Summary:**

This paper proposes a system for object pose estimation from RGB images. The only input to the system is the RGB image, it does not use depth maps or 3D models. The system first uses an object detector to generate feature maps and bounding boxes, which are passed to a transformer module, along with auxiliary information such as box center coordinates. The output of the transformer is then passed to a translation head and a rotation head, which together output a pose for each object present. The method performs comparably to methods that use 3D models such as PoseCNN, and outperforms other RGB-only methods. The authors also demonstrate that their method can be used for camera localization given landmarks of known ground truth poses.

**Issues:**

See weaknesses above

**Quality Of The Limitations Section:**

Limitations are addressed clearly

**Reviewer Expertise:**

4: The reviewer is confident but not absolutely certain that the evaluation is correct

**Robotics Focus:**

Highly relevant to robotics but no hardware experiments

**Strengths And Weaknesses:**

Strengths:
- The approach is RGB-only and does not require depth maps or 3D models of every object, so it is much more accessible as anyone with a regular camera could in theory use it

Weaknesses:
- While object pose estimation methods commonly struggle with accurate orientation prediction, the rotation error for PoET still seems a bit too high to truly be of use in robotic manipulation tasks
- The authors compare with methods that are several years old (PoseCNN and SilhoNet). Are they still the best methods to compare against?
- The method requires a large dataset of labeled images and is unlikely to generalize well to novel objects


**Summary Of Recommendation:**

This paper proposes an interesting new method for object pose estimation using transformers with a simple and elegant architecture, and the authors show that it outperforms baselines. However, I have reservations about the baselines as well as the usefulness of the method for robotics given the high rotation errors.

---

> ### Author Response · Authors · 2022-08-26
> **Reply to Reviewer v6yy**
>
> We are thankful to the reviewer for the points raised.
>
> * The main source of rotational error are objects with rotational symmetries around one or more axes (master chef can, tomato soup can, tuna fish can). For robotic manipulation tasks, uncertainties in rotation around a symmetry axis likely do not affect the outcome. In fact, our 2D-only competitors correct their rotational errors to remove the part stemming from rotational symmetries (see Table 3). We have investigated the axes of our rotation errors and compared them to the symmetry axes for symmetric objects. The average tilt of rotation error axes with respect to symmetry axes across all test images and symmetric objects is only 15 degrees. This confirms the fact that the largest part of our rotation error stems from rotations about the symmetry axes of symmetric objects.  If we ignore rotational errors about symmetry axes, our average rotation error reduces to 11.24 degrees, outperforming all RGB-only competitors (see Table 3). This analysis will be added to the manuscript. Additionally, we showed that the rotation estimation is sufficient for the camera localization task where these errors around the symmetry axis cannot be ignored (see Figure 2 and Table 4). In fact, our mean attitude error for the localized camera across all 12 test sequences is only 7.1, 12.4 and 19.6 degrees for roll, pitch, and yaw (see bb-all in Table 4), when using all objects in a single image for camera localization including symmetric objects and without any outlier rejection. To further motivate the use of PoET as a pose sensor in mobile robotics, we have integrated it in a state-of-the-art sensor fusion framework and performed state estimation experiments with YCB-V objects in our motion capture room using inertial data for propagation and PoET for the pose update. For a representative trajectory, the average error in rotation was (roll, pitch, yaw) = (4.0, 6.9, 14.8) degrees and in translation (x, y, z) = (0.084, 0.179, 0.032) meters. These results will be added to the manuscript. The performance for our real data is only slightly worse compared to the performance on the benchmark dataset, even though we were using a completely different camera, than the one used to record the original dataset, and the objects have slightly changed in appearance in comparison to the original YCB objects. Nonetheless, PoET is able to provide sufficiently accurate pose data.
> * While PoseCNN and SilhoNet may not be the newest methods, to the best of our knowledge, they are the only state-of-the-art methods that do not use any 3D model information for pose refinement (PNP, iterative refinement) and report on all the metrics that we use. We also compare our method to T6D, which is the most recent method that only evaluates on RGB-images, even though it requires 3D models for the loss calculation during training. There is no publicly available implementation and hence we can not compare our PoET to it for different metrics. However, PoET outperforms T6D for the ADD-S metric, does not require 3D models during training or inference and can be trained on top of any object detector, simplifying the adaptation to new use cases and objects.
> * Requiring a large database is true for all deep learning methods that are compared in this work. However, for any novel object we just need to retrain the transformer part of PoET on top of an object detector capable of detecting the object. All other methods require to be completely trained from end-to-end and most of them also need an accurate 3D model.

---

### Official Review · Reviewer_6Xj1 · 2022-08-04

**Originality:** Fair
**Technical Quality:** Good
**Clarity Of Presentation:** Fair
**Impact:** 2

**Recommendation:**

Weak Reject: I recommend rejecting the paper, but will not argue for my recommendation if the majority of other reviewers have a different opinion.

**Summary:**

This paper presents a model, called PoET, for single-image pose estimation. They feed the image to a backbone object detector (Scaled-YOLO-v4), which produces multi-scale features and bounding boxes. An attention-decoder model subsequently processes each detected object, using an architecture based on Deformable DETR. The model then predicts rotation and translation, using geodesic loss and L2 distance. They evaluate the model on the YCB-V dataset, where their method gets better performance than many baselines, including T6d-direct and SilhoNet. Ablations in the supplementary material compare different design decisions, such as different model size and rotation representations.

**Issues:**

As mentioned above, it would be helpful for the authors to provide more explanation for how the the technical approach is a significant advance over T6d-direct, provide additional ablations related to DETR, and describe future writing revisions.

**Quality Of The Limitations Section:**

Limitations are addressed clearly

**Reviewer Expertise:**

3: The reviewer is fairly confident that the evaluation is correct

**Robotics Focus:**

Relevant but unlikely to deploy to hardware in near future

**Strengths And Weaknesses:**

Strengths:

- The approach obtains good performance on YCB-V. It outperforms previous approaches that do not have 3D models available. It obtains similar performance to T6d-direction, without
- The approach is a natural combination of object detection methods and transformers. It is arguably simpler than many of the baselines.

Weaknesses:
- The paper paper is not written very clearly. In particular, the paper does not very clearly explain its contribution, and the ablations are not well organized.
- Only one qualitative result is shown, and it is in the supplementary material.
- I think that the ablations should be moved to the main paper, and presented more concisely. Given that the contribution is the architecture (and a combination of existing ideas), and the improvement over previous methods is not large, I think that these ablations are very important. It would also be helpful to see these expanded to cover some of the other key contributions, such as the design of the Deformable DETR module (which is one of the main novelties).
- The approach is a straightforward application of existing methods (object detection followed DETR-like attention and regression). In particular, it's similar to T6D-Direct (which requires a 3D model for its symmetry-aware loss).


**Summary Of Recommendation:**

While this method gets good results with a simple method, the approach is quite incremental, and the paper is not very clearly written. Given that both the approach and the results are incremental, I would have expected there to be significantly more analysis. I think that this paper is an interesting step towards simpler, transformer-based pose regression, but it would benefit from additional revisions.

---

> ### Author Response · Authors · 2022-08-26
> **Reply to Reviewer 6Xj1**
>
> We want to thank the reviewer for the feedback.
> * In particular, we agree that the ablation study is an essential part and thus we will summarize the extensive ablation study from the supplementary material and include the most important conclusions in the main paper. We will use this to further support our contributions as outlined in lines 53 onwards and to motivate our modifications to the transformer architecture. We will also improve the presentation of the ablation study in the appendix.
> * We are also happy to move the qualitative result into the main part of the manuscript and show additional qualitative results in the appendix.
> * We have introduced a variety of modifications to the standard DETR architecture that are responsible for the performance improvements. In particular, we have replaced learnable reference points and query embeddings with predicted bounding box information from the object detector. As our ablation study shows (see last three columns in Table 2 in Supplementary), these modifications are key to the performance improvement of our model. In fact, learning both query embeddings and reference points as in the standard architecture results in an AUC of ADD-S metric of only 42.0, less than half of what we achieve with our baseline model. These and other conclusions will be moved to the main part of the paper.
> * While there is a similarity to T6D, we show in our paper that PoET can be trained on top of any object detection framework, thus extending it for 6D pose estimation. In contrast to that, T6D requires the backbone and the pose estimator to be retrained from scratch every time. Moreover, as correctly mentioned by the reviewer, we do not require a 3D model, which is a major difference and advantage compared to T6D's approach. Yet, we outperform T6D with our method on important benchmark datasets. We would like to point out that our direct competitors in the RGB only class perform significantly worse than us: PoseCNN 75.9, SilhoNet 79.6, and MCN 75.1 vs PoET (our approach) 87.1. Despite only using 2D RGB data for training and inference, we also outperform or have only slightly inferior performance than state of the art models using 3D data (see Table 1). In our view, this constitutes a significant contribution as the need for 3D data creates constraints for the training data and increases the computational complexity during inference.

---

### Meta-Review · Area_Chair_ze8g · 2022-08-15

**Recommendation:** Accept (Poster)
**Confidence:** 4

**Metareview:**

Dear Authors,

Thank you for submitting your manuscript to CoRL. I'm happy to inform you that your paper is acceptable for publication. We have completed the review of your manuscript and a summary is appended below.
The reviewers have advised accepting your manuscript as a poster after improvement of the quality of the manuscript presentation based on the comments.
Please note it is crucial to incorporate all provided explanations from authors to convince the reviewers and recommended editing into the final manuscript.

Regards,

---

> ### Author Response · Authors · 2022-08-26
> **Summary of the response to the reviewers and addressing main issues**
>
> We thank the Meta-Reviewer for summarizing the most important aspects
>
> * In the response to the reviewers we have highlighted the novelties of our approach. We will revise our manuscript and add a summary of our ablation study to the main paper as it underlines the novelties of our approach. While there is a similarity to T6D, PoET has some major advantages in terms of required data, computational complexity and adaptability to new objects with respect to retraining the network.
> * With the important points raised by Reviewer 4, we will revise our manuscript for the camera ready version and improve the presentation.
> * Similarly with respect to the presentation of our paper, we will improve the typos and formatting issues for the camera ready version
> * The main source for rotational error are objects with rotational symmetries around one axis (master chef can, tomato soup can, tuna fish can). For robotic manipulation tasks, uncertainties in rotation around a symmetry axis likely do not affect the outcome. We have investigated the axes of our rotation errors and compared them to the symmetry axes for symmetric objects. The average tilt of rotation error axes with respect to symmetry axes across all test images and symmetric objects is only 15 degrees. If we ignore rotational errors about symmetry axes, our average rotation error reduces to 11.24 degrees, outperforming all RGB-only competitors (see Table. 3) We will add this analysis to the manuscript. Additionally, we recorded real data in our motion capture room with real YCB-V objects and fused the pose estimates of PoET with inertial measurement data in a mobile robotics sensor fusion framework. The results over a representative trajectory shows that PoET can be used in a real world scenario (see our response to Reviewer 2 for details). This experiment will be added to the manuscript.
> * While PoseCNN and SilhoNet may not be the newest methods, to the best of our knowledge, they are the only state-of-the-art methods that do not use any 3D model information for pose refinement (PNP, iterative refinement) and report on all the metrics that we use. Moreover, we compare PoET to T6D, which is the most recent method that evaluates only on 2D images, but uses 3D object information during training. We outperform T6D on the ADD-S metric and our approach can be trained on top of any object detector. However, a detailed comparison to T6D is not possible as no public implementation is available.